# EHDV-2 Infection Prevalence Varies in *Culicoides sonorensis* after Feeding on Infected White-Tailed Deer over the Course of Viremia

**DOI:** 10.3390/v11040371

**Published:** 2019-04-23

**Authors:** Sandra Y. Mendiola, Mary K. Mills, Elin Maki, Barbara S. Drolet, William C. Wilson, Roy Berghaus, David E. Stallknecht, Jonathan Breitenbach, D. Scott McVey, Mark G. Ruder

**Affiliations:** 1Southeastern Cooperative Wildlife Disease Study, Department of Population Health, College of Veterinary Medicine, University of Georgia, 589 D.W. Brooks Drive, Athens, GA 30602, USA; sandra.y.mendiola@gmail.com (S.Y.M.); dstall@uga.edu (D.E.S.); 2Department of Entomology, Kansas State University, 1603 Old Claflin Place, Manhattan, KS 66506, USA; marymi@usca.edu; 3Arthropod-Borne Animal Diseases Research Unit, USDA, Agricultural Research Service, 1515 College Avenue, Manhattan, KS 66506, USA; elin@k-state.edu (E.M.); barbara.drolet@ars.usda.gov (B.S.D.); William.wilson@ars.usda.gov (W.C.W.); oldswab@gmail.com (J.B.); scott.mcvey@ars.usda.gov (D.S.M.); 4Department of Population Health, College of Veterinary Medicine, University of Georgia, 220 College Station Road, Athens, GA 30602, USA; berghaus@uga.edu

**Keywords:** *Culicoides sonorensis*, epizootic hemorrhagic disease virus, hemorrhagic disease, *Odocoileus virginianus*, orbivirus, white-tailed deer

## Abstract

Epizootic hemorrhagic disease viruses (EHDVs) are arboviral pathogens of white-tailed deer and other wild and domestic ruminants in North America. Transmitted by various species of *Culicoides*, EHDVs circulate wherever competent vectors and susceptible ruminant host populations co-exist. The impact of variation in the level and duration of EHDV viremia in white-tailed deer (*Odocoileus virginianus*) on *Culicoides* infection prevalence is not well characterized. Here we examined how infection prevalence in a confirmed North American vector of EHDV-2 (*Culicoides sonorensis*) varies in response to fluctuations in deer viremia. To accomplish this, five white-tailed deer were experimentally infected with EHDV-2 and colonized *C. sonorensis* were allowed to feed on deer at 3, 5, 7, 10, 12, 14, 18, and 24 days post infection (dpi). Viremia profiles in deer were determined by virus isolation and titration at the same time points. Blood-fed *Culicoides* were assayed for virus after a 10-day incubation (27 °C) period. We found that increases in deer EHDV blood titers significantly increased both the likelihood that midges would successfully acquire EHDV and the proportion of midges that reached the titer threshold for transmission competence. Unexpectedly, we identified four infected midge samples (three individuals and one pool) after feeding on one deer 18 and 24 dpi, when viremia was no longer detectable by virus isolation. The ability of ruminants with low-titer viremia to serve as a source of EHDV for blood-feeding *Culicoides* should be explored further to better understand its potential epidemiological significance.

## 1. Introduction

Epizootic hemorrhagic disease virus (EHDV) (*Orbivirus*: Reoviridae), the causative agent of epizootic hemorrhagic disease (EHD), is transmitted by biting midges of the genus *Culicoides* to a wide range of wild and domestic ruminants. The EHDV serogroup is comprised of seven serotypes worldwide [1], three of which (EHDV-1, -2, and -6) are considered endemic in the United States of America (USA) [2]. Outbreaks of EHD range from localized, isolated events to explosive epidemics that span large geographic areas [3]. In endemic regions, EHDV is thought to be maintained in a *Culicoides* vector–ruminant host cycle [3]. In North America, *Culicoides sonorensis* is the only confirmed vector of EHDV [4,5], although other *Culicoides* species are likely involved in transmission [6,7], and white-tailed deer (WTD; *Odocoileus virginianus*) are the most severely affected ruminant hosts, suffering significant morbidity and mortality [8].

Clinical disease in WTD is highly variable, ranging from subclinical infection to peracute disease and death. Several factors contribute to the variation in clinical outcomes of EHDV infection in WTD, including the virulence of circulating EHDV strains, cross-protection between serotypes [9], innate resistance of specific host populations [10], and herd immunity [11]. The presence of endemic or epidemic disease patterns has emerged as a good predictor of clinical outcomes at the landscape level. Namely, deer in endemic zones are more likely to experience mild disease, or even subclinical infection, whereas deer in epidemic regions often experience severe disease with high case fatality rates [12]. The level and duration of EHDV viremia in WTD can vary widely between individuals. Peak EHDV blood titers occur fairly early in the course of infection, typically around 4 to 6 days post infection (dpi) [10,13], but prolonged viremia in WTD is also possible with virus detectable in the blood of infected animals up to 59 dpi [10].

The duration and titer of EHDV viremia in WTD is an important consideration when determining their potential to serve as a virus source for *Culicoides* vectors. In *Culicoides*, a variety of factors influence vector competence, which refers to the ability of a vector to acquire, maintain, and subsequently transmit a pathogen. Female *C. sonorensis* may ingest EHDV while blood-feeding from infected deer, but not all midges will go on to transmit the virus. Once ingested by a *Culicoides* vector, EHDV must overcome several barriers to infection [14]. The likelihood of a midge becoming infected with EHDV and being capable of transmitting the virus is influenced by numerous genetic and environmental factors, such as the amount of pathogen ingested, the insect’s immune response, and vector-pathogen genotype interactions [14,15,16]. The timing of blood-feeding by a *Culicoides* midge relative to the stage of EHDV infection in WTD influences the amount of virus ingested, or if virus is ingested at all. High titer viremias in WTD should lead to more virus in the blood meal, elevating the probability that midges will develop disseminated EHDV infections and become competent vectors. Thus, the timing of blood-feeding likely plays a role in overcoming barriers that prevent or constrain successful EHDV infection in midges.

It is unknown whether WTD with prolonged viremia, particularly those with low-titer infections, serve as a source of virus to feeding *Culicoides*. In previous studies, *Culicoides* feeding on EHDV-infected deer with low-titer viremias had low infection prevalence [6,17]. However, the significance of these infection prevalences and how they may vary over time remains poorly understood. In the current study, we investigated how the kinetics of EHDV-2 infection in WTD affect the infection prevalence of *C. sonorensis* and characterized how the infection prevalence varied over the course of viremia. With this information, we can determine the time frame during which viremic WTD are most infectious to *Culicoides* and begin to discern the epidemiological significance of deer with prolonged and low-titer EHDV viremias.

## 2. Materials and Methods

### 2.1. Animals and Culicoides

Six hand-raised white-tailed deer were obtained from the Whitehall Deer Research Facility (University of Georgia, Athens, GA, USA) and transported to the Large Animal Research Center (Kansas State University, Manhattan, KS, USA). The fawns were housed indoors and were seven months old at the time of inoculation. Laboratory-reared *C. sonorensis* from colonies maintained at the Arthropod-Borne Animal Diseases Research Unit (USDA, Manhattan, KS, USA) were used and were 3–4 days post-emergence at the time of feeding. All animal procedures were approved by the Institutional Animal Care and Use Committee at Kansas State University (protocol #3438).

### 2.2. Virus and Inoculum

The EHDV-2 isolate used for inoculation was originally isolated at the Southeastern Cooperative Wildlife Disease Study from the spleen of a free-ranging WTD (CC12-304) from Coffey County, Kansas, during a 2012 EHD outbreak. The virus was originally isolated on cattle pulmonary artery endothelial (CPAE) cells (American Type Culture Collection, Manassas, VA, USA), passaged once in baby hamster kidney (BHK) cells (ATCC), and then to CuVaW8A (CuVa) cells (*Culicoides sonorensis* cell line; USDA-ARS) [18,19]. The virus stock was 10^6.2^ tissue culture infective doses (TCID_50_)/mL as determined by virus titration using CPAE cells in a 96 well format as described in [13] and endpoint titers were determined [20]. Sham inoculum for negative control contained culture media from cell culture flasks not inoculated with virus.

### 2.3. Experimental Design

Five deer were inoculated with 2 mL of virus stock (10^6.5^ TCID_50_) by a combination of subcutaneous and intradermal injections (0.05–0.1 mL per injection) in the cervical and inguinal regions. The negative control deer was similarly administered a sham inoculum. Each animal was visually monitored for clinical signs of disease twice daily. At 0, 3, 5, 7, 10, 12, 14, 18, and 24 dpi, deer were sedated for physical examination, rectal temperature, blood collection, and *Culicoides* feeding. Deer were sedated with intramuscular xylazine (1–2 mg/kg; AnaSed, Lloyd Laboratories, Shenandoah, IA, USA) and sedation was reversed with slow intravenous injection of tolazoline (2–4 mg/kg; Lloyd Laboratories). Blood in sodium citrate was used for virus isolation and titration. Serum from additive-free blood tubes was used for serology. All EHDV-infected animals were euthanized at 24 dpi by intravenous injection of sodium pentobarbital (1 mL/5 kg).

Midges were allowed to feed on all five infected deer while under sedation on 3, 5, 7, 10, 12, 14, 18, and 24 dpi, using previously described protocols [17]. Briefly, cages containing 150–200 midges (male and female), were allowed to feed for 20–30 min on the skin of the ventral abdomen or inner thigh. After each feeding trial, midges were immobilized with carbon dioxide and sorted by blood-feeding status (i.e., blood-fed or non-blood-fed) under a dissecting microscope. After feeding, five blood-fed midges were placed into individual 1.5-mL microcentrifuge tubes containing 500 μL of virus transport media (minimum essential medium (MEM) with 10% fetal bovine serum (FBS)) and antibiotic/antimycotic solution (500 units penicillin, 0.5 mg streptomycin, and 1.25 μg amphotericin B/mL) (Sigma Chemical Company, St. Louis, MO, USA) to test whether virus could be isolated directly from the blood meals prior to any incubation period. All remaining blood-fed midges were held in an insectary for 10 days at 27 °C on a 12:12 light–dark cycle, and provided 10% sucrose ad libitum. Following incubation, all surviving midges were frozen individually or in pools (24 dpi midges only) in midge transport media and stored at −80 °C until processed for virus isolation.

### 2.4. Virology and Serology

#### 2.4.1. Deer

To evaluate the diagnostic utility of various cell lines for EHDV, virus isolation from deer blood was performed on CPAE cells, BHK cells, and CuVa cells. All attempts for virus isolation and titration from blood were performed on the day of collection using methods similar to those previously described [13]. Briefly, one mL of blood was washed in nine mL Dulbecco’s phosphate buffer saline (DPBS; Sigma) and centrifuged, with the wash removed and discarded. This wash step was repeated three times, and the one mL of washed erythrocytes were sonicated and centrifuged. Supernatant diluted 1:10 in MEM was used to inoculate (200 µL) monolayered CPAE cells, BHK cells, and CuVa cells in 6-well formats. Virus media used for CPAE cells consisted of MEM with 10% FBS and 1X antibiotics/antimycotic solution. Virus medium used for BHK cells was similar except 2% FBS was used. Medium used for CuVa cells was prepared as described previously in [18]. Both BHK and CPAE cells were incubated in 5% CO_2_ at 37 °C, whereas CuVa cells were incubated at 26 °C. After seven days, cultures were passaged by aspirating 100 uL of supernatant with cells from the wells and inoculating them onto fresh monolayers for a second passage. For CuVa cells, second passage was made onto CPAE cells. Cell culture supernatant was collected from cultures exhibiting cytopathic effect (CPE) and RNA extracted using a QIAamp Viral RNA Mini kit (Qiagen, Valencia, CA, USA) according to manufacturer instructions. Virus isolates were confirmed by RT-PCR using previously published primers [21]. Virus titration of blood samples was performed using CPAE cells in a 96 well format as described in [13] and endpoint titers were determined [20]. Serotype-specific antibodies were detected by serum neutralization, and antibody titers were determined as previously described in [11] except for the use of CPAE cells instead of BHK cells.

#### 2.4.2. Culicoides

Individual or pooled midges (5 midges/pool; 24 dpi only) were manually homogenized in microcentrifuge tubes using sterile pestles before being sonicated for 15 s in a sonicating water bath (Branson, Sonic Power Company, Danbury, CT, USA). Homogenized midges were then centrifuged at 4 °C for 12 min at 1500× *g*. For virus isolation, 100 µL of supernatant were inoculated onto BHK cell monolayers in a 24-well (individual midges) or 12-well plate format (pooled midges). Plates were incubated for 7 days in 5% CO_2_ at 34 °C and wells were monitored daily for CPE. After 7 days, samples were passaged by aspirating 100 µL of cells and supernatant from the plate and inoculating them onto fresh BHK monolayers for a second 7-day culture in 5% CO_2_ at 37 °C [22].

For all samples where CPE was observed on the second passage, the original sample was titrated for virus using BHK cells as previously described [22] and endpoint titers (TCID_50_/midge) were determined [20]. The minimum titer detectable through our assay was 10^2.3^ TCID_50_/midge. To calculate and graph mean virus titers, a value of 10^1.15^ TCID_50_/midge was assigned to samples that were CPE positive, but titer was below the limit of detection. Our approach assumes all values between 10^2.3^ TCID_50_/midge and zero could be present in these cases and their average titer could be as high as half of the limit of detection [23]. We considered midges with virus titers ≥10^2.7^ TCID_50_/midge to be potentially competent as previous work with bluetongue virus (BTV)—an orbivirus closely related to EHDV—has demonstrated that *C. sonorensis* midges reaching these titers are capable of efficient virus transmission to susceptible hosts [24]. Further, this threshold was substantiated for EHDV-2 where a titer of ≥10^2.7^ TCID_50_/midge corresponded with the dissemination of virus from the mid-gut to organs including the salivary glands [25].

### 2.5. Statistics

Log_10_ EHDV titers were compared over time using linear mixed models with deer as a random effect. The percentage of midges with positive virus isolation results, and with titers 10^2.7^ TCID_50_, were compared over time using mixed logistic regression models with deer as a random effect. Pairwise comparisons of different time points were performed using the Bonferroni procedure to limit the type I error probability to 5% over all comparisons. All tests assumed a two-sided alternative hypothesis, and *p* < 0.05 was considered statistically significant. Analyses were performed using commercially available statistical software (Stata version 14.2, StataCorp LP, College Station, TX, USA).

## 3. Results

### 3.1. WTD Infection

All five inoculated deer had a detectable viremia and developed mild to moderate clinical disease. Mild clinical signs were observed in deer 2, 3, 4, and 6, and included elevated body temperature, hyperemia of oral mucosa and conjunctiva, and erythema of non-haired regions: ear pinnae, perineum, nares, periorbital zones. In addition to the above clinical signs, deer 5 exhibited moderate clinical disease, which included depression, lethargy, and lameness. The lameness observed in deer 5 occurred at 11–15 dpi and was characterized by a reluctance to stand and a hunched posture with rigid forelimb extension while attempting to walk. Small hemorrhages in the oral mucosa were occasionally observed on 10 dpi (deer 3, 5, and 6). The baseline body temperature was 39.1 °C, determined by –3 and 0 dpi recordings for all deer in the study. Peak elevations in body temperature occurred on 5 dpi for deer 6 (+1.2 °C) and 7 dpi for deer 2 (+0.2 °C), 3 (+0.8 °C), 4 (+1.3 °C), and 5 (+0.9 °C). Viremia profiles for individual deer are presented in Figure 1 and the mean is presented in Figure 2. Using CPAE data, three of five (60%) deer in the study had a detectable EHDV-2 viremia by 3 dpi and all were viremic by 5 dpi. Peak viremias were observed either at 5 or 7 dpi for all deer. Mean viremia for the 5 deer peaked at 10^3.98^ TCID_50_/_mL_ at 5 dpi (Table 1) while peak viremias in individual deer ranged from 10^3.7^ to 10^6^ TCID_50_/_mL_ (Figure 1). There was a significant difference in deer titers over time (*p* < 0.001). Compared to 3, 5, and 7 dpi, the mean titers on 12 and 14 dpi were significantly lower. The mean titer on 10 dpi was intermediate, and did not differ significantly from the mean titer on any other day. EHDV-2 was not detected in any deer after 14 dpi.

Virus isolation results varied over the course of infection depending on the cell line used (Figure 3). There was 100% agreement on the three cell lines at 5 and 7 dpi, when blood virus titer was highest in all five deer. However, there was discrepancy later in the course of viremia (10–14 dpi). The CPAE cells yielded positive virus isolation results on four time points that were virus isolation negative using CuVa cells and eight time points that were virus isolation negative using BHK cells. The CuVa cells yielded virus positive results at one time point that was virus negative using CPAE.

Neutralizing antibodies were first detected in all deer on day 7 and all had an antibody titer of 160 or higher by 12 dpi. Neutralizing antibodies were not detected at any time point in the sham inoculated negative control.

### 3.2. C. sonorensis Infection

Midge infection prevalence, calculated as the number of midges that were virus isolation positive out of the total number of midges tested, varied over the course of viremia and between individual deer in the trial. Virus was rarely isolated from midges that were sampled prior to the 10-day incubation period. We successfully isolated virus from 6 out of 265 (2.3%) of these midges (five from deer 5 at 7 dpi and one from deer 3 at 7 dpi). All six cases resulted in titers below the threshold of detection. All analyses omitted these midges and focused solely on midges that underwent the 10-day incubation period. Midge virus isolation results and titers are summarized in Table 1 and the distribution of deer titers relative to virus isolation results in midges are illustrated in Figure 1 and Figure 2. Midge data from deer 5, 5 dpi, and deer 4, 7 dpi, were omitted from all calculations and graphs due to contamination that prevented accurate virus isolation and titration values.

There was a significant difference in the proportion of midges with positive virus isolation results over time (*p* < 0.001). Compared to 5 dpi, the proportion of midges with positive virus isolation results was significantly lower at 3, 10, 14, and 18 dpi. EHDV-2 was not detected in midges at 12 dpi. There was also a significant relationship between the percentage of midges with a positive virus isolation result and deer EHDV titers (*p* < 0.001). For every one-log_10_ increase in a deer’s EHDV titer, the odds that midges feeding on that deer would have a positive virus isolation result were 2.6 times higher (95% CI: 2.1, 3.3).

Days 5 and 7 post infection produced the most virus positive midges, accounting for 69 out of 85 (81.2%) of the total infected midges processed over the course of the study. In addition to midge infection prevalence derived from virus isolation data, we determined midge viral titers and the proportion of infected midges at each time point that reached titers ≥ 10^2.7^ TCID_50_ indicating they could serve as potential vectors for EHDV-2 (Figure 4). Mean virus titers for positive midges differed significantly over time (*p* < 0.001) and ranged from 10^1.15^ TCID_50_/midge to 10^3.54^ TCID_50_/midge. There was also a significant difference in the proportion of midges with an EHDV titer ≥ 10^2.7^ TCID_50_ over time (*p* < 0.001). High percentages of infected midges from 5 and 7 dpi were determined to be potentially competent vectors, 68.2% and 80%, respectively (Figure 4). In fact, only midges that fed from deer at 5 and 7 dpi were found to have mean titers that were higher than the threshold of transmission competence (Figure 5). Mean midge titers for all other days fell below the limit of detection (10^2.3^ TCID_50_/midge). Mean titers on 5 and 7 dpi were significantly higher than the mean titer at 14 dpi, but no other pairwise comparisons between days were statistically significant (Table 1). Mean midge titers could not be calculated for 12 dpi because EHDV was not detected in any midges on that day. Compared to 5 dpi, the proportion of midges with a titer ≥10^2.7^ TCID_50_ was significantly lower at 3 and 10 dpi, and was not statistically different on 7 dpi. None of the sampled midges had a titer ≥10^2.7^ TCID_50_ on 12, 14, or 18 dpi. There was a significant relationship between midge EHDV titers and deer EHDV titers (*p* < 0.001). For every one-log_10_ increase in a deer’s EHDV titer, the EHDV titer in midges feeding on that deer increased by an average of 0.49 log_10_ units (95% CI: 0.26, 0.73). Likewise, there was a statistically significant correlation between the percentage of midges with an EHDV titer ≥ 10^2.7^ TCID_50_ and deer EHDV titers (*P* < 0.001). For every one-log_10_ increase in a deer’s EHDV titer, the odds that midges feeding on that deer would have an EHDV titer ≥ 10^2.7^ TCID_50_ were 3.7 times higher (95% CI: 2.5, 5.4).

Although no virus was isolated from the blood of any deer on 18 dpi, we were able to isolate viruses from 3 midges that fed on deer 5 at 18 dpi and from a pool of midges (*n* = 5) that fed on deer 5 at 24 dpi. Incidentally, deer 5 was the deer that reached the highest EHDV-2 titer (peak 10^6^ TCID_50_/_mL_) in our trial.

## 4. Discussion

Clinical signs of EHD observed in this study were generally mild. However, moderate clinical disease was observed in deer 5, with prominent lethargy and lameness present 11–15 dpi. Interestingly, this animal had the highest peak blood virus titer, and there is often a correlation between peak viremia and severity of clinical disease during experimental infections. An unexpected feature of the observed viremia profiles was the relatively short duration. While significant individual variation exists, deer infected with EHDV often have a prolonged viremia lasting as long as 59 days [10]. All deer in this study had a detectable viremia at 14 dpi, but blood samples from all five deer were virus isolation negative by 18 dpi. The reason for this apparent short viremia is not known, but may be related to the strain of the virus inoculum, attenuation of a cell culture-derived inoculum, or the distribution of inoculum between cervical and inguinal regions. A secondary objective of this study was to compare various cell culture lines for EHDV isolation from deer blood. *Culicoides* cells (KC cells) have previously been shown to outperform other culture systems [25], although CPAE cells had not been evaluated. Here, we included CPAE cells in our evaluation because this cell line has been relied upon for regional orbivirus diagnostics for many years [26]. Viremia was detected one time point later in three of five deer using CPAE cells, confirming their diagnostic utility for EHDV in white-tailed deer.

Our results indicate that EHDV-2 midge infection prevalence tracks fluctuations in white-tailed deer viremia. We found that increases in the blood EHDV titer of deer significantly increased the likelihood that midges would successfully acquire EHDV, as measured through positive virus isolation. Despite the variation in viremia profiles between individual deer, the highest proportion of virus-positive midges observed for each deer tended to coincide with peak viremia, typically 5 to 7 dpi, and correspondingly decreased as blood virus titer decreased as deer recovered from infection.

To evaluate the potential for EHDV-positive midges to transmit EHDV, we also determined the virus titers of individual positive midges. Once ingested by a midge, the virus must overcome a variety of barriers to infect and eventually escape the midgut and disseminate to the salivary glands [15]. While the titer of midges capable of EHDV transmission has not been established, we considered *C. sonorensis* with virus titers ≥ 10^2.7^ TCID_50_/midge potentially competent vectors, a value extrapolated from BTV studies in domestic sheep that showed *C. sonorensis* reaching titers of ≥ 10^2.7^ TCID_50_/midge were capable of infecting susceptible hosts [24]. Because of the various factors underlying individual midge susceptibility to EHDV, only a portion of *C. sonorensis* were expected to achieve vector competence at any given time point examined [14]. Our results indicate that the proportion of EHDV-2-competent *C. sonorensis* is strongly influenced by the viral load of the deer they feed on. As expected, the percentages of competent *C. sonorensis* peaked with peak blood virus titer in deer, with more than half of EHDV-2-infected midges considered competent at 5 and 7 dpi. Not only were the highest infection prevalences observed on these days, but large percentages of those infected midges were potentially competent. No other time point yielded as many infected or potentially competent *C. sonorensis* in comparison. In fact, no potentially competent midges were found after 10 dpi at which time there was only one potentially competent midge. These results suggest that WTD are the most infectious to *C. sonorensis* during peak viremia, relatively soon after first becoming infected.

While we were able to isolate EHDV-2 from midges that fed on deer as far along as 18 and 24 dpi, in the case of deer 5, these infection prevalences were very low. Previous work involving the infection of *C. sonorensis* with EHDV has produced similar results, with deer viremias <10^4.0^ TCID_50_/_mL_ typically resulting in low infection prevalence [12,17] and titers < 10^2.3^ TCID_50_/_mL_ yielding no infected midges [25]. It was common for the deer in our study to have blood virus titers below these thresholds, especially at later time points. Our study shows that EHDV-2 infection prevalence in *C. sonorensis* varies over the course of viremia in WTD as a response to fluctuations in viral titer. Midges are most likely to acquire EHDV-2 and become competent vectors when feeding on deer at peak viremias. Though not conclusive, our results indicate that feeding on EHDV-infected deer with low blood virus titers results in inefficient host-to-vector transmission of EHDV-2.

The ability of WTD to sustain low-titer EHDV viremias for prolonged periods suggests these animals are a potential source of EHDV for feeding *Culicoides*. The recovery of virus from *Culicoides* after feeding on deer 5 on 18 and 24 dpi was unexpected, as this animal did not have a detectable viremia by virus isolation at these time points. However, similar findings have occurred in sheep infected with BTV, a closely related orbivirus [25]. The ability of ruminants with low-titer viremia to serve as a source of EHDV for feeding *Culicoides* is an interesting finding and should be explored further to better understand its potential significance. Considering the low number of virus positive midges detected at these time later time points, our findings suggest that low titer EHDV viremias in WTD result in extremely inefficient host-to-vector transmission. However, attack rates on white-tailed deer by *Culicoides* have been shown to be extremely high in nature, with one study regularly recovering >10,000 *C. debilipalpis* per morning from individual deer during peak midge season [22]. Thus, even low infection rates may yield several hundred potentially infectious midges in the wild. Additional studies are required for a more accurate estimate of EHDV host-vector transmission efficiency at low titers.

## Figures and Tables

**Figure 1 viruses-11-00371-f001:**
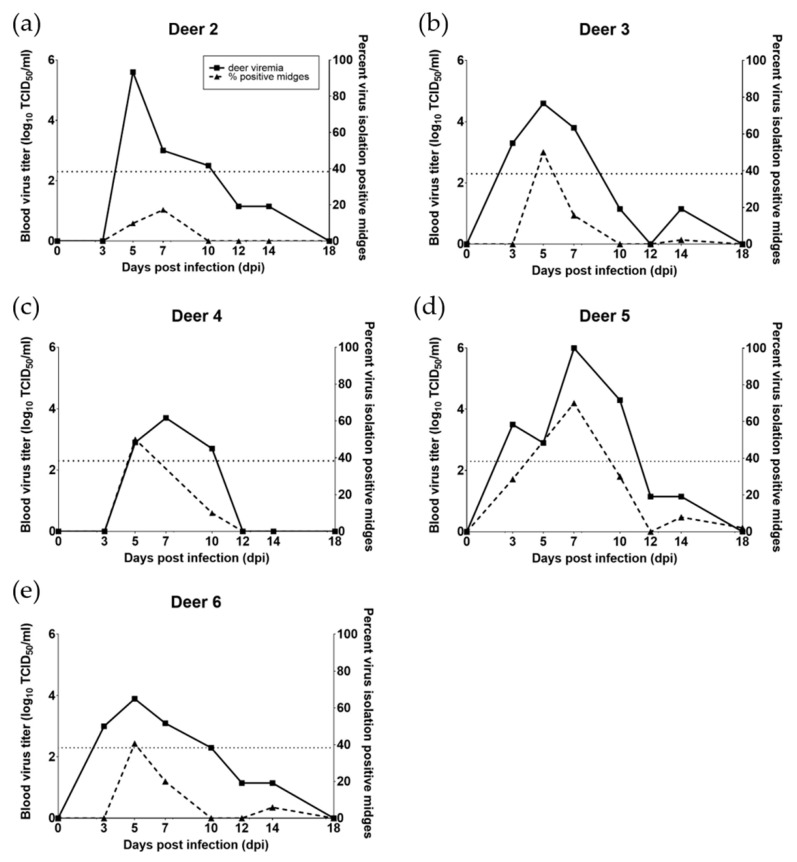
Viremia profiles spanning from 0 days post infection (dpi) to 18 dpi are plotted alongside midge infection prevalence for individual deer in the trial (**a**–**e**). A dotted line at 2.3 on the primary y-axis of each graph indicates the limit of detection for viral titer. Midge infection data for deer 5, 5 dpi and deer 4, 7 dpi have been omitted from their respective graphs due to contamination of samples during virus isolation.

**Figure 2 viruses-11-00371-f002:**
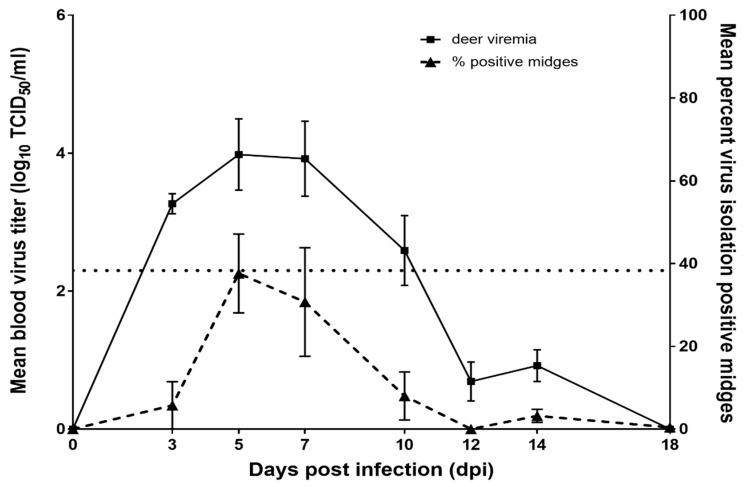
Mean epizootic hemorrhagic disease virus serotype 2 titer of viremic deer and proportion of infected midges determined by pooling data from all 5 deer for each time point. Standard error bars are shown for both mean deer viremia and midge infection prevalence. A lack of standard error bars indicates no variation in values observed. The dotted line at 2.3 on the primary y-axis indicates the limit of detection for blood virus titer (10^2.3^ tissue culture infective dose (TCID)_50_/mL).

**Figure 3 viruses-11-00371-f003:**
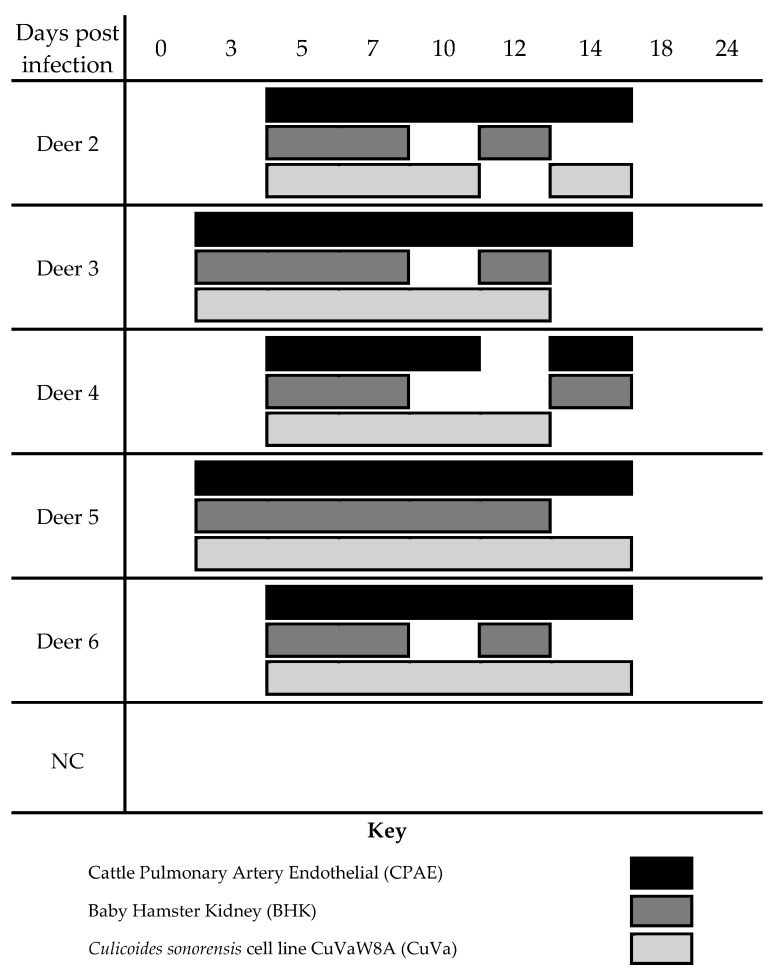
Evaluation of cell lines for EHDV diagnostics. Virus isolation from blood of deer was attempted on three different cell lines at different time points throughout the study. Shaded cells indicate positive virus isolation.

**Figure 4 viruses-11-00371-f004:**
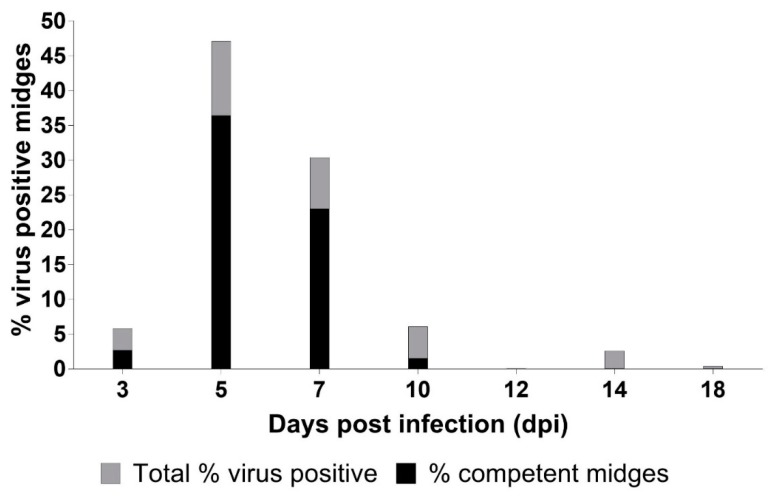
Bar graph showing proportion of epizootic hemorrhagic disease virus serotype 2 positive midges that reached titers > 10^2.7^ median tissue culture infective doses (TCID_50_)/midge and were considered competent to transmit virus. No positive midges were detected at 12 dpi.

**Figure 5 viruses-11-00371-f005:**
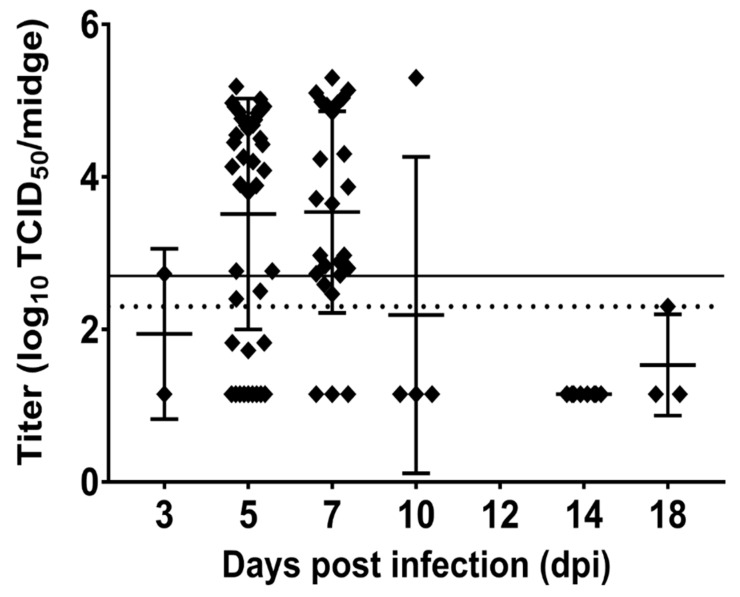
Plot showing titration values from all EHDV-2 positive midges. Each point on the graph represents a single midge. Horizontal bars indicate the mean titer while vertical bars reflect standard deviation. On the y-axis, the solid line at 2.7 denotes the threshold of transmission competence (10^2.7^ TCID_50_/midge) and the dotted line at 2.3 denotes the limit of detection (10^2.3^ TCID_50_/midge).

**Table 1 viruses-11-00371-t001:** Summary of virus isolation results by day post infection (dpi) for midges that fed from five deer experimentally infected with epizootic hemorrhagic disease virus (EHDV) serotype 2.

dpi	Mean (SE) log_10_ Titer for Deer with Viremia	Number of Midges Evaluated	Percent (SE) of Midges Virus Isolation Positive	Percent (SE) of Midges with Titer ≥10^2.7^ TCID_50_ /Midge	Mean (SE) Titer for Virus Isolation Positive Midges
3	3.27 ^b^ (0.50)	30	5.8 ^a,b^ (4.9)	2.7 ^a^ (3.1)	1.94 ^a,b^ (0.99)
5	3.98 ^b^ (0.39)	127	47.1 ^c^ (10.9)	36.4 ^b^ (11.0)	3.51 ^b^ (0.21)
7	3.92 ^b^ (0.39)	86	30.4 ^b,c^ (10.2)	23.0 ^a,b^ (9.3)	3.54 ^b^ (0.28)
10	2.59 ^a,b^ (0.39)	85	6.1 ^a,b^ (4.1)	1.5 ^a^ (1.8)	2.19 ^a,b^ (0.70)
12	1.15 ^a^ (0.39)	157	^2^ 0.0	^3^ 0.0	^1^ ND
14	1.15 ^a^ (0.39)	373	2.6 ^a^ (1.8)	^3^ 0.0	1.15 ^a^ (0.53)
18	^1^ ND	498	0.4 ^a^ (0.4)	^3^ 0.0	1.53 ^a,b^ (0.81)

Within the columns, estimated marginal means and percentages with a superscript in common do not differ with a level of significance of 5% over all comparisons. ^1^ ND = not determined. No titer calculated because there were no virus-positive samples. ^2^ EHD virus was not detected in any midges on these days necessitating their exclusion from the mixed logistic regression model. ^3^ No midges had a detectable EHDV titer ≥10^2.7^ TCID_50_/midge on these days.

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
