# Peer review of "EHDV-2 Infection Prevalence Varies in Culicoides sonorensis after Feeding on Infected White-Tailed Deer over the Course of Viremia"

_viruses, 2019, doi:10.3390/v11040371_

Reviewer 1 Report

 In their manuscript “EHDV-2 infection prevalence varies in Culicoides sonorensis after feeding on infected white-tailed deer 3 over the course of viremia” Mendiola et al report on their results of an experiment examining the relationship between EHDV titer in experimentally infected deer and vectors allowed to feed upon the deer at several time points (up to 24 days). Considering the difficulty of performing this work, the authors deserve much credit.  The Introduction, Methods and Results sections are well written and clear. The Discussion is rather weak, and only perfunctorily attempts to interpret the significance of the work (which is substantial). For example, the authors state that “The reason for this apparent short viremia is not known, but may be related to a variety of host or virus factors.” yet they do not go on to discuss these “factors” not speculate on why their specific deer had short viremia (compared to >50 days in some studies). Another example is the statement that “The ability of WTD to sustain low-titer EHDV viremias for prolonged periods has led manyto question the role of these animals as sources of EHDV for Culicoides” that does not provide citations of who the many are or what they think may be going on, if deer are not the source of virus. What are prevailing alternate maintenance hypotheses? How do the authors’ findings support or refute those hypotheses? This is important work and the authors should be telling us how their results help to answer outstanding questions.

Another quibble is the use of macerated whole midges for the vector infection parts. The authors did not collect midge saliva to show that the virus was present and transmissible by bite. At a minimum this should be discussed (with citations and summary of other studies) as a caveat or limitation, in the Discussion section.

TITLE

The finding suggested by the title “EHDV-2 infection prevalence varies in Culicoides sonorensis after feeding on infected white-tailed deer 3 over the course of viremia” is not explicitly expressed in the abstract.  Either the title or abstract should be altered so that these “match”

 ABSTRACT

Line 20: Delete “well-known”.

Line 33: Provide some details on “low numbers” 

Line 35: Delete “ is an interesting finding and”

 INTRODUCTION

Line 46: The statement “EHDV is maintained through a Culicoides vector–ruminant host cycle” does not have citations and may not be accurate. In the upper Midwest, for example, where prolonged (>6 month) winters limit vector activity for long periods, other mechanisms of maintenanceshould exist. 

To clarify the authors could state that “EHDV is transmittedin a Culicoides vector–ruminant host cycle, although alternate hosts may be involved in virus maintenance during inter-epizootic periods”.

 Lines 71-72: Add “should” after “High titer viremias in WTD” as no reference is provided to substantiate the claim.

 Line 77: Change “have” to “had” to mathc tense with “previous studies” (or say “were found to have”).

 MATERIALS AND METHODS

Line 86: Delete comma after “Six”.

Line 88: Please insert (in parentheses) the age cohort of the deer after “and were seven months old” to help the reader. Are they fawns, subadults, teenagers? 

Line 90: Replace “old” with “post-emergence”.

Line 92: Insert IACUC protocol number.

Line 101: Need period at end of sentence.

Lines 117-118. Please state why “five blood-fed midges were placed into individual 1.5-ml 117 microcentrifuge tubes containing 500-μl of virus transport media”

 FIGURES

Font sizes in figure 1 are quite small. Should be doubled in size.

Figure 1 caption: The lengthy explanation of  the axes and lines are not needed. Delete from “Viral titer” to “deer 6”.

 Figure 2 caption: Delete the sentence beginning “Viral titer”

 Figure 4: What does “VI” refer to in the legend?

 RESULTS

Line 241: Change “was statistically similar on 7 dpi” to “was not statistically different on 7 dpi”

 Line 247: Delete the sentence “The distribution of EHDV-2 titers for individual midges is illustrated in Figure 5.” and refer to Figure 5 in the following sentence.

 The paragraphs starting on lines 233 and 246 seem redundant. I recognize that they present different metrics, but they are presented in a very similar fashion. Could these paragraphs be combined to compare and contrast these 2 measures of infection?

 DISCUSSION

Lines 270-271: I find the explanation that “The reason for this apparent short viremia is not known, but may be related to a variety of host or virus factors.” to be too vague and in need of more detailed discussion of previous work. All of these deer were the same age (young). Is there a age-class driven pattern of viremic period in WTD? What are other factors that could explain the short term viremia in the current work and its discrepancy with other studies?

Author Response

The comments by the reviewers are appreciated. Below are all of the reviewer comments in black, followed by our response and edits in blue. We are happy to make additional changes to the manuscript at the request of the assistant editor.

 Point-by-point response to reviewers

 TITLE

 The finding suggested by the title “EHDV-2 infection prevalence varies in

Culicoides sonorensis after feeding on infected white-tailed deer 3 over the

course of viremia” is not explicitly expressed in the abstract. Either the title

or abstract should be altered so that these “match”

 Given space limitations of abstracts, it seems redundant to have to restate the title in the abstract. If the AE would like this done, please advise.

 ABSTRACT

Line 20: Delete “well-known”.

 Change incorporated

Original: “...are well-known arboviral pathogens...”

Revised: “...are arboviral pathogens...”

 Line 33: Provide some details on “low numbers”

 Change incorporated

Original: “...we identified low numbers...”

Revised: “...we identified four infected midge samples (three individuals and one pool)...”

 Line 35: Delete “is an interesting finding and”

 Change incorporated

Original: “...Culicoides is an interesting finding and should be explored further to ...”

Revised: “...Culicoides should be explored further to...”

 INTRODUCTION

Line 46: The statement “EHDV is maintained through a Culicoides vector–

ruminant host cycle” does not have citations and may not be accurate. In the

upper Midwest, for example, where prolonged (>6 month) winters limit vector

activity for long periods, other mechanisms of maintenance should exist.

To clarify the authors could state that “EHDV is transmitted in

a Culicoides vector–ruminant host cycle, although alternate hosts may be

involved in virus maintenance during inter-epizootic periods”.

 This is true, there are potentially unique mechanisms at play to describe this but they remain largely unexplored and are speculative. Another possibility for northern detections is the repeated introduction of viruses. In California, some have shown persistence of virus in culicoides during cooler periods, so there are many potential mechansims at play that may vary regionally. I’ve made the following change to reflect what occurs in endemic regions. I prefer to not get too speculative

 Change incorporated

Original: “...maintained through a Culicoides vector–ruminant host cycle...”

Revised: “In endemic regions, EHDV is thought to be maintained in a Culicoides vector–ruminant host cycle [3].”

 Lines 71-72: Add “should” after “High titer viremias in WTD” as no reference

is provided to substantiate the claim.

 Change incorporated

Original: “...viremias in WTD lead...”

Revised: “...viremias in WTD should lead…”

 Line 77: Change “have” to “had” to match tense with “previous studies” (or

say “were found to have”).

 Change incorporated

Original: “...low-titer viremias have low infection...”

Revised: “...low-titer viremias had low infection…”

 MATERIALS AND METHODS

Line 86: Delete comma after “Six”.

 Change incorporated

Original: “...Six, hand-raised white-tailed deer...”

Revised: “...Six hand-raised white-tailed deer…”

 Line 88: Please insert (in parentheses) the age cohort of the deer after “and

were seven months old” to help the reader. Are they fawns, subadults,

teenagers?

 Change incorporated

Original: “The deer were housed indoors and were seven months old at the time of inoculation.”

Revised: “The fawns were housed…”

 Line 90: Replace “old” with “post-emergence”.

 Change incorporated

Original: “...were used and were 3-4 days old...”

Revised: “...were used and were 3-4 days post-emergence…”

 Line 92: Insert IACUC protocol number.

 Change incorporated

 Line 101: Need period at end of sentence.

 Change incorporated

Original: “...not inoculated with virus”

Revised: “...not inoculated with virus.”

 Lines 117-118. Please state why “five blood-fed midges were placed into

individual 1.5-ml 117 microcentrifuge tubes containing 500-ÎĽl of virus

transport media”

 Original: “… solution [500 units penicillin, 0.5 mg streptomycin, and 1.25 ÎĽg amphotericin B/ml] (Sigma Chemical Company, St. Louis, MO, USA) for virus isolation.”

 Revised: “… solution [500 units penicillin, 0.5 mg streptomycin, and 1.25 ÎĽg amphotericin B/ml] (Sigma Chemical Company, St. Louis, MO, USA) to test whether virus could be isolated directly from the blood meals prior to the extrinsic incubation period.”

 We further address this in the results through the following addition:

 L236-240

Virus was rarely isolated from midges that were sampled prior to the 10-day incubation period. We successfully isolated virus from 6 out of 265 (2.3%) of these midges (five from deer 5 at 7 dpi and one from deer 3 at 7 dpi). All six cases resulted in titers below the threshold of detection. All analyses omitted these midges and focused solely on midges that underwent the 10-day incubation period.  

 FIGURES

Font sizes in figure 1 are quite small. Should be doubled in size.

 Font sizes were adjusted accordingly, additionally, Figure 1 was restructured to make every graph larger for easier visualization.

 Figure 1 caption: The lengthy explanation of the axes and lines are not

needed. Delete from “Viral titer” to “deer 6”.

  Change incorporated

Original: “Viremia profiles spanning from 0 dpi to 18 dpi are plotted alongside midge infection rates for individual deer in the trial. Viral titer is plotted on the primary y-axis and percentage of midges that were virus isolation positive is plotted on the secondary y-axis for (a) deer 2, (b) deer 3, (c) deer 4, (d) deer 5, and (e) deer 6. A dotted line at 2.3 on the primary y-axis of each graph indicates the limit of detection for viral titer…”

 Revised: “Viremia profiles spanning from 0 dpi to 18 dpi are plotted alongside midge infection rates for individual deer in the trial. A dotted line at 2.3 on the primary y-axis of each graph indicates the limit of detection for viral titer.”

 Figure 2 caption: Delete the sentence beginning “Viral titer”

 Change made as recommended.

  Figure 4: What does “VI” refer to in the legend?

 We replaced “VI” which stood for virus isolation with virus positive in the figure 4 legend for continuity with the graph axes and enhanced clarity.

 RESULTS

Line 241: Change “was statistically similar on 7 dpi” to “was not statistically

different on 7 dpi”

 Change incorporated

Original: “and was statistically similar on 7 dpi.”

Revised: “and was not statistically different on 7 dpi.”

Line 247: Delete the sentence “The distribution of EHDV-2 titers for individual

midges is illustrated in Figure 5.” and refer to Figure 5 in the following

sentence.

 Change incorporated

 The paragraphs starting on lines 233 and 246 seem redundant. I recognize

that they present different metrics, but they are presented in a very similar

fashion. Could these paragraphs be combined to compare and contrast

these 2 measures of infection?

              Change incorporated: paragraphs merged & better integrated

Original: Days 5 and 7 post infection produced the most positive midges, accounting for 69 out of 85 (81.2%) of the total infected midges processed over the course of the study. In addition to midge infection rates derived from virus isolation data, we determined the proportion of infected midges at each time point that reached titers ≥ 102.7 TCID50 indicating they could serve as potential vectors for EHDV-2 (Figure 4). There was a significant difference in the proportion of midges with an EHDV titer ≥ 102.7 TCID50 over time (P < 0.001). High percentages of infected midges from 5 and 7 dpi were determined to be potentially transmission competent vectors, 68.2% and 80%, respectively (Figure 4). Compared to 5 dpi, the proportion of midges with a titer ≥ 102.7 TCID50 was significantly lower at 3 and 10 dpi, and was statistically similar on 7 dpi. None of the sampled midges had a titer ≥ 102.7 TCID50 on 12, 14, or 18 dpi. There was also a statistically significant correlation between the percentage of midges with an EHDV titer ≥ 102.7 TCID50 and deer EHDV titers (P < 0.001). For every one-log10 increase in a deer’s EHDV titer, the odds that midges feeding on that deer would have an EHDV titer ≥ 102.7 TCID50 were 3.7 times higher (95% CI: 2.5, 5.4).

Mean virus titers for positive midges differed significantly over time (P < 0.001) and ranged from 101.15 TCID50/midge to 103.54 TCID50/midge. The distribution of EHDV-2 titers for individual midges is illustrated in Figure 5. Only midges that fed from deer at 5 and 7 dpi were found to have mean titers that were higher than the threshold of transmission competence. Mean midge titers for all other days fell below the limit of detection (log10 2.3 TCID50/midge). Mean titers on 5 and 7 dpi were also significantly higher than the mean titer at 14 dpi. No other pairwise comparisons between days were statistically significant (Table 1). Mean midge titers could not be calculated for 12 dpi because EHDV was not detected in any midges on that day. There was also a significant relationship between midge EHDV titers and deer EHDV titers (P < 0.001). For every one-log10 increase in a deer’s EHDV titer, the EHDV titer in midges feeding on that deer increased by an average of 0.49 log10 units (95% CI: 0.26, 0.73).

 Revised:  Days 5 and 7 post infection produced the most virus positive midges, accounting for 69 out of 85 (81.2%) of the total infected midges processed over the course of the study. In addition to midge infection rates derived from virus isolation data, we determined midge viral titers and the proportion of infected midges at each time point that reached titers ≥ 102.7 TCID50 indicating they could serve as potential vectors for EHDV-2 (Figure 4). Mean virus titers for positive midges differed significantly over time (P < 0.001) and ranged from 101.15 TCID50/midge to 103.54 TCID50/midge. There was also a significant difference in the proportion of midges with an EHDV titer ≥ 102.7 TCID50 over time (P < 0.001). High percentages of infected midges from 5 and 7 dpi were determined to be potentially competent vectors, 68.2% and 80%, respectively (Figure 4). In fact, only midges that fed from deer at 5 and 7 dpi were found to have mean titers that were higher than the threshold of transmission competence (Figure 5). Mean midge titers for all other days fell below the limit of detection (log10 2.3 TCID50/midge). Mean titers on 5 and 7 dpi were significantly higher than the mean titer at 14 dpi, but no other pairwise comparisons between days were statistically significant (Table 1). Mean midge titers could not be calculated for 12 dpi because EHDV was not detected in any midges on that day. Compared to 5 dpi, the proportion of midges with a titer ≥ 102.7 TCID50 was significantly lower at 3 and 10 dpi, and was not statistically different on 7 dpi. None of the sampled midges had a titer ≥ 102.7 TCID50 on 12, 14, or 18 dpi. There was a significant relationship between midge EHDV titers and deer EHDV titers (P < 0.001). For every one-log10 increase in a deer’s EHDV titer, the EHDV titer in midges feeding on that deer increased by an average of 0.49 log10 units (95% CI: 0.26, 0.73).Likewise, there was a statistically significant correlation between the percentage of midges with an EHDV titer ≥ 102.7 TCID50 and deer EHDV titers (P < 0.001). For every one-log10 increase in a deer’s EHDV titer, the odds that midges feeding on that deer would have an EHDV titer ≥ 102.7 TCID50 were 3.7 times higher (95% CI: 2.5, 5.4).

 DISCUSSION

Lines 270-271: I find the explanation that “The reason for this apparent short

viremia is not known, but may be related to a variety of host or virus factors.”

to be too vague and in need of more detailed discussion of previous work. All

of these deer were the same age (young). Is there a age-class driven pattern

of viremic period in WTD? What are other factors that could explain the short

term viremia in the current work and its discrepancy with other studies?

 Change incorporated

Sentence now reads: “The reason for this apparent short viremia is not known, but may be related to the strain of the virus inoculum, attenuation of a cell culture-derived inoculum, distribution of inoculum between cervical and inguinal regions.”

 We can’t explain the shorter viremia and there are many potential explanations for this. All explanations come with a lot of speculation. However, fawns from this research herd and this age have been used in many EHDV studies prior to this one and I don’t suspect a problem there. The strain of virus, cell culture attenuation, inoculation method, could all be at play but we can’t say for sure. I can elaborate on this if the AE prefers.

Reviewer 2 Report

General comments

 This paper by Mendiola et al. describes the relationships between viremia profile of white-tailed deer infected with EHDV-2 and the infection prevalence of Culicoides sonorensis after feeding. As the authors mentioned, it has been mostly unknown how viremia profiles in the EHDV-infected ruminants affect the following transmission by the bite of Culicoides vectors. To clarify this matter, the authors investigated viremia profiles of the EHDV-infected deer, EHDV prevalence of Culicoides sonorensis after feeding on the deer, and the EHDV titer in the infected midges quite carefully and properly. This study is well-designed, and it contains lots of valuable data that would be of interest for the readers. Particularly, it is interesting that they identified infected midges after feeding on a deer 18- and 24- days post infection, although the viremia in the deer was undetectable level. In addition, they used three cell lines for virus isolation, CPAE, BHK and CuVa, so the information shown in this paper would be useful for diagnosis of EHD. Experimental infections of arboviruses using natural hosts and vectors usually require quite lots of time, labor and costs, but the authors accomplished this experimental EHDV-2 infection of white-tailed deer with the vector of EHDV in North America, Culicoides sonorensis. I express my respect for the authors’ great efforts.

I would like to give some minor comments as follows.

 #1  P. 3, line 95

The information about the EHDV-2 isolate used in this study should be described (reference about the isolate and/or GenBank accession number of the isolate).

 #2  P. 3, lines 104-105

How did you determine the titer of the virus and the routes of infection? Maybe it is good to describe the reason for the titer and routes in the paper.

 #3  P. 3, lines 114-115

I understand that the conditions of feeding are described in Ref #17, but the information about the midges, such as the number of the midges per deer, sex, duration of the time for feeding, should be described in this paper.

Author Response

REVIEWER 2

 #1 P. 3, line 95

The information about the EHDV-2 isolate used in this study should be

described (reference about the isolate and/or GenBank accession number of

the isolate).

 Included the SCWDS accession number:

“The EHDV-2 isolate used for inoculation was originally isolated at the Southeastern Cooperative Wildlife Disease Study from the spleen of a free-ranging WTD (CC12-304) from Coffey County, Kansas, during a 2012 EHD outbreak.”

 #2 P. 3, lines 104-105

How did you determine the titer of the virus and the routes of infection?

Maybe it is good to describe the reason for the titer and routes in the paper.

 The sentence now reads: “The virus stock was 6.2 log10 tissue culture infective doses (TCID50)/ml as determined by virus titration using CPAE cells in a 96 well format as described [13] and endpoint titers were determined [20].”

 Inoculation route described in the subsequent paragraph.

 #3 P. 3, lines 114-115

I understand that the conditions of feeding are described in Ref #17, but the

information about the midges, such as the number of the midges per deer, sex, duration of the time of feeding, should be described in this paper.

 The following sentence was added:

“Briefly, cages containing 150-200 midges (male and female), were allowed to feed for 20-30 minutes on the skin of the ventral abdomen or inner thigh.”